# Impact of bisoprolol and amlodipine on cardiopulmonary responses and symptoms during exercise in patients with chronic obstructive pulmonary disease

**Chou-Chin Lan**[1,2], **Po-Chun Hsieh**[3], **I-Shiang Tzeng**[4], **Mei-Chen Yang**[1,2], **Chih-Wei Wu**[1,2], **Wen-Lin Su**[1,2], **Yao-Kuang Wu**[1,2]*

**1** Department of Internal Medicine, Division of Pulmonary Medicine, Taipei Tzu Chi Hospital, Buddhist Tzu Chi Medical Foundation, New Taipei City, Taiwan, Republic of China, **2** School of Medicine, Tzu-Chi University, Hualien, Taiwan, Republic of China, **3** Department of Chinese Medicine, Taipei Tzu Chi Hospital, Buddhist Tzu Chi Medical Foundation; School of Post-Baccalaureate Chinese Medicine, Tzu Chi University, Hualien, Taiwan, Republic of China, **4** Department of Research, Taipei Tzu Chi Hospital, Buddhist Tzu Chi Medical Foundation, New Taipei City, Taiwan, Republic of China

* drbfci@gmail.com

## Abstract

### Background

Patients with chronic obstructive pulmonary disease (COPD) often have exercise intolerance. The prevalence of hypertension in COPD patients ranges from 39–51%, and β-blockers and amlodipine are commonly used drugs for these patients.

### Objectives

We aimed to study the impact of β-blockers and amlodipine on cardiopulmonary responses during exercise.

### Methods

A total 81 patients with COPD were included and the patients underwent spirometry, cardiopulmonary exercise tests, and symptoms questionnaires.

### Results

There were 14 patients who took bisoprolol and 67 patients who did not. Patients with COPD taking ß-blockers had lower blood oxygen concentration ($SpO_2$) and more leg fatigue at peak exercise but similar exercise capacity as compared with patients not taking bisoprolol. There were 18 patients treated with amlodipine and 63 patients without amlodipine. Patients taking amlodipine had higher body weight, lower blood pressure at rest, and lower respiratory rates during peak exercise than those not taking amlodipine. Other cardiopulmonary parameters, such as workload, oxygen consumption at peak exercise, tidal volume at rest or exercise, cardiac index at rest or exercise were not significantly different between

**Data Availability Statement:** All relevant data are within the paper and its Supporting Information files.

**Funding:** This study was supported by grants from the Taipei Tzu Chi Hospital (TCRD-TPE-111-RT-3 (1/3)) and the Buddhist Tzu Chi Medical Foundation (TCMF-JCT 111-14). The funders had no role in study design, data collection and analysis, decision to publish, or preparation of the manuscript.

**Competing interests:** The authors have declared that no competing interests exist.

patients with or without bisoprolol or amlodipine. Smoking status did not differ between patients with or without bisoprolol or amlodipine.

## Conclusions

COPD is often accompanied by hypertension, and β-blockers and amlodipine are commonly used antihypertensive drugs for these patients. Patients with COPD taking bisoprolol had lower $SpO_2$ and more leg fatigue during peak exercise. Patients taking amlodipine had lower respiratory rates during exercise than those not taking amlodipine. Exercise capacity, tidal volume, and cardiac index during exercise were similar between patients with and without bisoprolol or amlodipine.

## Introduction

Chronic obstructive pulmonary disease (COPD) has a great impact on global health, and COPD is the third leading cause of death [1]. The incidence is increasing due to persistent noxious substance exposure and aging processes [2]. Patients with COPD often experience cough, shortness of breath, wheezing, poor health-related quality of life (HRQL), and poor exercise capacity [1]. COPD and hypertension frequently coexist in the same patient. The prevalence of hypertension in COPD patients ranges from 39% to 51%, and increases in older patients [3]. β-blockers and amlodipine are common drugs for patients with COPD and hypertension.

ß-blockers are recommended not only for hypertension, but also for heart diseases, such as heart failure, myocardial infarction, and angina [4]. ß-blockers are known to be beneficial in reducing mortality from heart diseases [4]. However, early studies revealed decreased lung function in patients with COPD taking β-blockers [5,6]. At that time, the use of β-blockers in patients with COPD was not encouraged due to concerns about inducing bronchospasm and possible adverse reactions. However, many recent studies have shown that β-blockers are well tolerated in patients with COPD without evidence of worsening lung function or respiratory symptoms [4,7]. Salpeter et al. conducted the first meta-analysis that showed that ß-blockers did not impair lung function in patients with COPD [7]. Cardioselective β-blockers are recommended when appropriate for patients with COPD [4]. However, the prevalence of using β-blockers in COPD patients remains low.

Amlodipine is a dihydropyridine calcium-channel blocker that inhibits the transmembrane influx of calcium into cardiac and vascular smooth muscles [8]. The antihypertensive effect of amlodipine is due to peripheral vasodilation and a reduction in systemic vascular resistance. Amlodipine not only controls hypertension but also reduces adverse cardiovascular events [9]. Once-daily amlodipine provides good blood pressure (BP) control and is generally well-tolerated [8]. Therefore, amlodipine is one of the most commonly prescribed antihypertensive drugs [8].

β-blockers and amlodipine are common drugs for patients with COPD and hypertension. Although, there have been some studies using ß-blockers or amlodipine on lung function in patients with COPD, studies on the effect of ß-blockers or amlodipine on cardiorespiratory responses during exercise still limited. Therefore, we conducted this study to comprehensively assess the effects of ß-blockers (bisoprolol) and amlodipine on cardiopulmonary responses and symptoms during exercise in patients with COPD.

## Materials and methods

### Study design and patient recruitment

Patients with COPD were retrospectively recruited from the out-patient department of Taipei Tzu-Chi Hospital. The diagnosis and severity of COPD were defined according to the Global Initiative for Chronic Obstructive Lung Disease (GOLD) guidelines [10]. The inclusion criteria were stable COPD with absence of acute exacerbation (AE) for at least 3 months before exercise tests, full ability to perform the exercise test, and willingness to be included in the exercise tests. The exclusion criteria were inability to perform exercise tests, such as orthopedic or neurological impairment, and history of other lung diseases (such as pneumoconiosis, tuberculosis, and asthma). Patients with a history of documented heart disease, such as congestive heart failure or coronary heart disease, were excluded to prevent the influence of heart disease on cardiopulmonary response to the cardiopulmonary exercise test (CPET). A total of 81 COPD patients who met the inclusion and exclusion criteria and completed CPET. To analyze the effects of β-blockers, patients were divided into groups based on whether they took bisoprolol (n = 14) or not (n = 67). To analyze the effects of amlodipine, patients were divided into groups based on whether they took amlodipine (n = 18) or not (n = 63).

This study was approved by the ethics committee of Taipei Tzu-Chi Hospital. Informed consent was obtained from all participants. All the patients underwent spirometry, CPET, respiratory muscle strength testing, and symptom evaluation during maximal exercise.

### Pulmonary function test

A spirometer (Medical Graphics Corporation, St Paul, MN, USA) was used to assess pulmonary function, according to the recommendations of the American Thoracic Society [11]. The diagnosis and severity of airflow obstruction were based on GOLD guidelines [10].

### Cardiopulmonary exercise test

The CPET was performed using a bicycle ergometer (Lode Corival, Netherlands). An incremental protocol was performed for CPET. Exhaled air was analyzed using breath analysis (Breeze Suite 6.1; Medical Graphics Corporation, St. Paul, MN, USA) to assess oxygen uptake ($VO_2$), carbon dioxide output ($VCO_2$), respiratory rate (RR), and tidal volume ($V_T$), systolic BP (SBP), diastolic BP (DBP), heart rate (HR), and oxygen saturation of the blood ($SpO_2$) were monitored during CPET. Cardiac index (CI) was calculated according to a previous study in which CI during exercise was $(4.3 \times VO_2 + 4.5)$/body surface area (BSA) in men and $(4.9 \times VO_2 + 3.6)$/BSA in women [12]. Pressure rate product (RPP), equal to HR multiplied by SBP, is an indicator of myocardial oxygen demand [13]. RPP was calculated at rest and peak exercise to define myocardial workload. Work efficiency (WE) is defined as the slope of $VO_2$/ work rate (WR), and it is determined using linear regression analysis [14]. $VO_2$ at the anaerobic threshold (AT) was determined using the V-slope method ($VO_2$ versus $VCO_2$ graph) [15]. Oxygen pulse ($O_2P$) was defined as $VO_2$ divided by HR ($O_2P = VO_2/HR$) [16]. Maximal WR and $VO_2$ at peak exercise ($VO_2$peak) are defined as exercise capacity [16].

### Respiratory muscle strength

A pressure gauge (Respiratory Pressure Meter; Micro Medical Corp, England) was used to measure maximum inspiratory pressure (MIP) and maximum expiratory pressure (MEP). MIP was measured when patients exhaled to the residual volume and subsequently performed a rapid maximal inspiration [17]. MEP was measured when the patients inhaled the total lung capacity and then exhaled with maximal effort.

### Symptoms score at maximal exercise

Dyspnea and leg fatigue scores were evaluated using the Borg scale with 10-point scores at rest and maximal exercise. Higher scores indicated more severe symptoms [18].

### Health-related quality of life (HRQL)

The HRQL was assessed using the Chinese version of COPD assessment test (CAT) that was provided by the Taiwan Society of Pulmonary and Critical Care Medicine. CAT contains eight items (cough, phlegm, chest tightness, breathlessness, limited activities, confidence leaving home, sleeplessness, and energy), each scored on a scale of 0–5 [19]. The total score ranges from 0 to 40 points, with higher scores indicating worse HRQL [19].

The modified medical research council (mMRC) dyspnea scale has five statements that describe the degree of dyspnea, from no dyspnea (grade 0) to almost complete incapacitation (grade 4) [20]. A mMRC $\geq$ 2 indicates significant dyspnea [20].

**Severe acute exacerbation (AE).** AE of COPD is a sudden worsening of symptoms in patients with COPD. Severe AE is defined as AEs requiring urgent medical attention, such as emergency department visits or hospitalizations [21]. We recorded the number of severe AEs in patients in the year preceding the analysis.

### Statistical analysis

All parameters were presented as the mean ± standard deviation. Statistical analyses were performed using Statistical Product and Service Solutions version 24.0 (SPSS, Inc., Chicago, IL, USA). An independent sample t-test was used to compare parameters between patients with and without bisoprolol or with and without amlodipine. Pearson's chi-square test was used to compare gender, smoking status, inhaled and oral medication between groups. The threshold for statistical significance was set at $P < 0.05$.

## Results

### Baseline clinical and demographic characteristics

The clinical characteristics of the patients are summarized in Table 1. The mean age was 60.8 ± 8.9 years, the mean body weight (BW) was 66.0 ± 14.3 kg, the mean body height (BH) was 164.1 ± 8.9 cm. 69 (85%) patients with male and 12 (15%) patients were female. 18 patients (22%) were never smoking, 45 patients (56%) were ex-smoking and 18 patients (22%) were current smoking. The mean smoking amount were 21.3 ± 19.9 pack years. Most patients were COPD GOLD stage II or III. The mean forced expiratory volume in the first one second (FEV1)/ forced vital capacity (FVC) was 60.8 ± 8.9%, FVC was 4.70 ± 2.46 L (78.7 ± 19.8%), and FEV1 was 3.22 ± 1.52 L/min (62.5 ± 20.9%). Age, BH, gender, and smoking status did not differ between patients with and without bisoprolol (p>0.05) or with and without amlodipine (p>0.05). However, the BW (71.0±7.1 kg) of patients with amlodipine was higher than the BW (64.6±15.6 kg) of patients without amlodipine (p = 0.015). The inhaled medications, oral procaterol and theophylline did not differ in patients with or without bisoprolol or amlodipine (p>0.05). However, patients taking oral theophylline were more in patients with amlodipine than those without amlodipine (p<0.05).

### Exercise capacity in patients using ß-blockers or amlodipine

The parameters of exercise capacity (WR and VO$_2$peak) and VO$_2$ at rest are shown in Table 2. The WR and VO$_2$ at rest and peak exercise were not significantly different between patients with and without bisoprolol or with and without amlodipine (all p>0.05).

**Table 1. Baseline characteristics.**

| | All (n = 81) | Bisoprolol | | | Amlodipine | | |
|---|---|---|---|---|---|---|---|
| | | No use (N = 67) | Use (N = 14) | P value | No use (N = 63) | Use (N = 18) | P value |
| Age (years) | 60.8 ± 8.9 | 67.7 ± 7.3 | 64.6 ± 7.6 | 0.165 | 67.0±7.6 | 67.6±6.7 | 0.758 |
| BH (cm) | 164.1 ± 8.9 | 164.4±9.3 | 162.6±6.7 | 0.504 | 163.5±9.2 | 166.0±7.6 | 0.296 |
| BW (Kg) | 66.0 ± 14.3 | 66.1±14.9 | 65.4±11.6 | 0.866 | 64.6±15.6 | 71.0±7.1 | 0.015 |
| COPD GOLD stages | | | | | | | |
| I | 18 | 13 | 5 | 0.587 | 11 | 7 | 0.360 |
| II | 41 | 35 | 6 | | 13 | 8 | |
| III | 21 | 18 | 3 | | 19 | 2 | |
| IV | 1 | 1 | 0 | | 0 | 1 | |
| Gender (male/female) | 69/12 | 58/9 | 11/3 | 0.427 | 54/9 | 15/3 | 0.802 |
| Smoking status | | | | | | | |
| Never smoking | 18 (22%) | 15 (22%) | 3 (21%) | 0.816 | 17 (27%) | 1 (5%) | 0.116 |
| Ex-smoking | 18 (22%) | 14 (21%) | 4 (29%) | | 12 (19%) | 6 (33%) | |
| Current smoking | 45 (56%) | 38 (57%) | 7 (50%) | | 34 (53%) | 11 (62%) | |
| Smoking (pack-years) | 21.3 ± 19.9 | 21.6 ± 20.6 | 20.0 ± 16.5 | 0.785 | 21.1 ± 21.5 | 21.9 ± 13.5 | 0.884 |
| Inhaled medication | | | | 0.148 | | | 0.198 |
| LAMA+LABA+ICS | 56 | 43 | 13 | | 40 | 16 | |
| LAMA+LABA | 17 | 17 | 0 | | 15 | 2 | |
| LABA+ICS | 6 | 5 | 1 | | 6 | 0 | |
| LAMA | 2 | 2 | 0 | | 2 | 0 | |
| Procaterol (yes/no) | 4/77 | 4/63 | 0/14 | 0.348 | 4/59 | 0/18 | 0.273 |
| Theophylline (yes/no) | 52/29 | 43/24 | 9/5 | 0.994 | 35/28 | 17/1 | 0.002 |

Abbreviations: BH, body height; BW, body weight; COPD, chronic obstructive pulmonary disease; GOLD, Global Initiative for Chronic Obstructive Lung Disease; ICS, inhaled corticosteroids; LABA, long-acting β2-agonists; LAMA, long-acting muscarinic antagonists.

## Cardiovascular response in patients using ß-blockers or amlodipine

The parameters of cardiovascular responses during exercise are shown in Table 3. SBP and DBP at rest were lower in patients taking amlodipine than in those not taking amlodipine (both p<0.05). The WE, $O_2P$, $VO_2$ at AT, SBP and DBP at exercise, cardiac index (CI) at rest and exercise, RPP at rest and exercise were not significantly different between patients with and without bisoprolol or with and without amlodipine (all p>0.05).

**Table 2. Antihypertensive drugs and exercise capacity.**

| | No use/ use | WR watt | $VO_2$, rest mL/min | $VO_2$, exercise mL/min | $VO_2$, exercise mL/min/kg | $VO_2$, exercise % |
|---|---|---|---|---|---|---|
| Bisoprolol | 67 | 80.1±35.7 | 283.3±93.9 | 1,121.1±390.2 | 18.3±10.5 | 71.3±23.1 |
| | 14 | 85.5±23.7 | 299.6±94.3 | 1,177.4±316.9 | 18.5±5.4 | 72.6±19.6 |
| | p value | 0.489 | 0.557 | 0.615 | 0.926 | 0.851 |
| Amlodipine | 63 | 78.5±32.4 | 278.1±97.4 | 1,123.9±371.7 | 18.9±10.6 | 73.1±20.6 |
| | 18 | 89.9±38.4 | 314.5±74.7 | 1,154.7±406.6 | 16.2±5.4 | 66.2±27.7 |
| | p value | 0.211 | 0.147 | 0.762 | 0.295 | 0.250 |

Abbreviations: $VO_2$, oxygen uptake; WR, work rate.

**Table 3. Antihypertensive drugs and cardiovascular parameters.**

| | No use/ | WE | AT | O$_2$P | HR, exercise | SBP, rest | DBP, rest | SBP, exercise | DBP, exercise | CI, rest | CI, exercise | RPP, rest | RPP, exercise |
|---|---|---|---|---|---|---|---|---|---|---|---|---|---|
| | use | mL/min/watt | mL/min | mL/beats | BPM | mmHg | mmHg | mmHg | mmHg | L/min/m2 | L/min/m2 | mmHg*BPM | mmHg*BPM |
| Bisoprolol | 67 | 8.50±1.54 | 743.1±170.9 | 9.10±2.91 | 121.9±16.7 | 123.4±17.1 | 75.3±11.5 | 171.1±29.7 | 80.5±13.0 | 5.23±0.50 | 9.69±2.08 | 9609±2089 | 21047±5490 |
| | 14 | 8.97±1.28 | 700.4±150.6 | 9.93±2.97 | 123.9±26.8 | 121.1±18.8 | 74.1±9.4 | 169.9±30.6 | 77.1±6.1 | 5.29±0.52 | 10.2±1.81 | 9723±2403 | 20975±5451 |
| | p value | 0.286 | 0.393 | 0.340 | 0.726 | 0.645 | 0.720 | 0.892 | 0.144 | 0.657 | 0.446 | 0.857 | 0.965 |
| Amlodipine | 63 | 8.78±1.38 | 729.3±170.8 | 9.19±2.98 | 122.7±18.8 | 120.2±16.9 | 74.0±11.8 | 169.9±29.9 | 79.9±12.5 | 5.20±0.52 | 9.70±1.99 | 9466±2038 | 20964±5393 |
| | 18 | 7.86±1.73 | 755.6±156.7 | 9.44±2.77 | 120.8±18.3 | 132.8±15.1 | 78.9±7.5 | 174.3±29.6 | 80.2±11.1 | 5.37±0.41 | 10.0±2.2 | 10198±2403 | 21280±5795 |
| | p value | 0.055 | 0.580 | 0.747 | 0.704 | 0.006 | 0.040 | 0.583 | 0.932 | 0.201 | 0.573 | 0.201 | 0.830 |

Abbreviations: AT, aerobic threshold; BPM, beats per minute; CI, cardiac index; DBP, diastolic blood pressure; HR, heart rate; O$_2$P, oxygen pulse; RPP, Rate Pressure Product; SBP, systolic blood pressure; WE, work efficiency.

## Ventilatory response in patients using ß-blockers or amlodipine

The parameters of the ventilatory responses during exercise are presented in Table 4. Patients with bisoprolol had lower SpO$_2$ (92.8±5.3) during exercise than SpO$_2$ (94.6±1.5) during exercise in patients without ß-blockers (p = 0.018). FEV1/FVC, FEV1, FVC, RR at rest or exercise, V$_T$ at rest or exercise, and SpO$_2$ at rest were not significantly different between patients with and without bisoprolol (all p>0.05). Patients taking amlodipine had a lower RR during exercise than patients not taking amlodipine (p<0.05). FEV1/FVC, FEV1, FVC, RR at rest, V$_T$ at rest or exercise, and SpO$_2$ at rest and exercise were not significantly different between patients with and without amlodipine (all p>0.05).

**Table 4. Antihypertensive drugs and ventilatory parameters.**

| | No use/ | FEV1/FVC | FVC | FVC | FEV1 | FEV1 | RR, rest | RR, exercise | V$_T$, rest | V$_T$, exercise | SpO$_2$, rest | SpO$_2$, exercise | MIP | MEP |
|---|---|---|---|---|---|---|---|---|---|---|---|---|---|---|
| | use | % | L | % | L/min | % | Breaths/min | Breaths/min | ml | ml | % | % | cmH2O | cmH2O |
| Bisoprolol | 67 | 60.5±8.9 | 2.44±0.70 | 78.0±20.1 | 1.51±0.55 | 61.9±21.1 | 19.1±7.5 | 33.0±7.8 | 683.1±224.9 | 1248.6±407.8 | 95.2±2.4 | 92.8±5.3 | 75.3±28.4 | 119.2±30.6 |
| | 14 | 62.1±8.8 | 2.53±0.59 | 82.3±18.8 | 1.59±0.50 | 65.4±20.7 | 20.8±5.5 | 34.4±6.8 | 563.7±114.4 | 1231.1±280.7 | 95.9±1.5 | 94.6±1.5 | 65.5±26.9 | 109.6±38.8 |
| | p value | 0.565 | 0.656 | 0.465 | 0.593 | 0.567 | 0.421 | 0.534 | 0.580 | 0.878 | 0.256 | 0.018 | 0.246 | 0.314 |
| Amlodipine | 63 | 60.2±7.9 | 2.41±0.72 | 76.9±19.4 | 1.48±0.55 | 60.6±19.8 | 19.9±7.5 | 34.3±7.6 | 648.1±220.2 | 1210.7±400.0 | 95.5±2.4 | 93.2±5.4 | 72.4±27.2 | 115.1±30.6 |
| | 18 | 62.7±11.7 | 2.64±0.52 | 85.2±20.4 | 1.68±0.50 | 69.0±23.8 | 17.6±5.7 | 29.7±6.7 | 712.8±189.3 | 1367.9±319.7 | 94.7±1.5 | 92.9±3.0 | 77.8±31.9 | 126.2±36.6 |
| | p value | 0.314 | 0.314 | 0.116 | 0.167 | 0.135 | 0.241 | 0.024 | 0.261 | 0.130 | 0.185 | 0.862 | 0.472 | 0.196 |

Abbreviations: FEV1, forced expiratory volume in 1 s; FVC, forced vital capacity; MEP, maximal expiratory pressure; MIP, maximal inspiratory pressure; RR, respiratory rate; SpO$_2$, peripheral capillary oxygen saturation; VEQ, ventilatory equivalents; V$_T$, tidal volume.

### HRQL and symptoms at peat exercise in patients using ß-blockers or amlodipine

The parameters of HRQL and symptoms during the peat exercise are shown in Table 5. Patients taking bisoprolol had a higher leg fatigue score at peak exercise than those not taking bisoprolol (p<0.05). The CAT score, mMRC, dyspnea Borg score at peak exercise, and severe AE were not significantly different between patients with and without bisoprolol (all p>0.05). For amlodipine, CAT, mMRC, dyspnea Borg score, leg fatigue Borg score, and severe AE were not significantly different between the patients with and without amlodipine (all p>0.05).

## Discussion

The effects of β-blockers and amlodipine on cardiopulmonary responses and symptoms during exercise in patients with COPD is an important issue. This study had several important findings. First, patients with COPD taking ß-blockers had lower $SpO_2$ and more leg fatigue at peak exercise but similar exercise capacity compared to patients not taking ß-blockers. Second, patients taking amlodipine had higher BW, lower BP at rest, and lower RR at peak exercise than those not taking amlodipine. Most cardiopulmonary parameters, such as WR, $VO_2$peak, FEV1, FVC, $V_T$ at rest or exercise, CI at rest or exercise, and HRQL, were not significantly different in patients with or without ß-blockers or amlodipine. However, most of the patients in the current study had moderate to severe COPD, so the findings are more appropriate for such patients.

In the current study, there were no significant differences in lung function, HRQL, or AE in patients with or without bisoprolol. Based on a recent meta-analysis, β-blocker use was associated with a reduction in AE in patients with COPD [4]. However, Maltais et al. conducted an analysis of a large population of 557 of 5,162 patients who received β-blockers, and AE was similar in these two groups [22]. The effects of β-blockers on AE in patients with COPD seem inconclusive. Previous studies have also shown that HRQL did not differ between patients with and without β-blockers assessed by the St. George's Respiratory Questionnaire (SGRQ) [22] or 36-Item Short Form Survey [23], which was similar to our study. Maltais et al. also showed that there were no differences in lung function and SGRQ scores between the β-blocker and nonβ-blocker groups [22].

Therefore, there are only few studies on the effects of β-blockers on cardiopulmonary responses during exercise in patients with COPD [24,25]. However, these studies did not reveal detailed parameters of cardiorespiratory exercise responses. In the current study, we revealed the comprehensive parameters of cardiopulmonary responses to exercise. We found no significant differences in exercise capacity (WR or $VO_2$peak) in patients taking bisoprolol or amlodipine. A previous study also showed that the exercise capacity measured using 6-min walking distance was not significantly different between patients with or without β-blockers [24]. Thirapatarapong et al. showed that patients with COPD with and without β-blockers showed similar $VO_2$peak [25]. These findings indicate that β-blockers do not adversely affect functional capacity in patients with COPD [25,26].

However, we found a decrease in $SpO_2$ at peak exercise in patients taking bisoprolol. Dynamic hyperinflation during exercise is important in patients with COPD. Mainguy et al. examined the effects of β-blockers on dynamic hyperinflation [27]. They showed that bisoprolol significantly worsened dynamic hyperinflation during exercise in patients with moderate-to-severe COPD without affecting lung function at rest [27]. Therefore, we suspected that β-blockers reduce airway caliber by affecting airway smooth muscle, which may have little effect on resting lung function, but they may still cause dynamic hyperinflation during exercise and result in a mild decrease in $SpO_2$. However, the magnitude of these effects was small, with only

**Table 5. Antihypertensive drugs and quality of life.**

|  | No use/ use | CAT | mMRC | Borg, dyspnea | Borg, leg fatigue | Severe AE |
|---|---|---|---|---|---|---|
| Bisoprolol | 67 | 12.3±5.2 | 0.19±0.53 | 3.83±1.39 | 2.93±1.58 | 0.16±0.45 |
|  | 14 | 12.1±2.8 | 0.36±0.63 | 4.58±1.98 | 4.67±2.46 | 0.43±0.65 |
|  | p value | 0.905 | 0.314 | 0.114 | 0.002 | 0.164 |
| Amlodipine | 63 | 12.1±4.1 | 0.22±0.55 | 3.96±1.59 | 3.37±1.95 | 0.21±0.48 |
|  | 18 | 12.9±7.0 | 0.22±0.55 | 3.93±1.16 | 2.53±1.09 | 0.22±0.55 |
|  | p value | 0.659 | 1.000 | 0.956 | 0.103 | 0.905 |

Abbreviations: AE, acute exacerbation; CAT, COPD assessment test; mMRC, modified. Medical Research Council.

a small decrease in SpO$_2$ and no effect on the exercise capacity. We suggest that the use of bisoprolol should not be contraindicated in patients with COPD.

In the current study, we found that patients taking bisoprolol had more prominent leg fatigue during exercise. A previous study also suggested that β-blockers may reduce the ability to deliver oxygen to muscles by causing muscle fatigue in the extremities [28]. Although, we found more leg fatigue during exercise, exercise capacity was similar in patients with or without bisoprolol. Therefore, patients taking bisoprolol had more leg fatigue, but without decreases in exercise capacity. A previous study also showed that β-blockers did not show meaningful differences in muscle strength [28] and exercise capacity [25]. Many patients with COPD are current ex-smoker or current smoker. Smoking is one of the most important contributors to peripheral arterial disease (PAD) [29]. The mechanisms of smoking-induced PAD are endothelial cell dysfunction, smooth muscle cell remodeling, and atherosclerotic vascular changes [29]. PAD may be associated with leg pain and limited physical function [29]. In our study, we did not investigate the prevalence of PAD in these patients. Therefore, the impact of PAD cannot be defined. However, there was no significant difference in smoking status between patients with or without bisoprolol.

Approximately, 20–90% of patients with COPD are reported to have secondary pulmonary hypertension (SPH) and cor pulmonale [30]. The increase in pulmonary arterial pressure (PAP) and pulmonary vascular resistance (PVR) have been shown to be associated with mortality in COPD [30]. As vasoconstriction might play a role in the pathogenesis of SPH in COPD, amlodipine might be a therapeutic choice in this patient population [31]. In a study of 20 patients with clinically stable COPD and SPH, amlodipine decreased PVR and PAP and improved right heart function [31]. Another study showed that plasma N-terminal pro-B-type natriuretic peptide (NT-ProBNP) levels were significantly reduced in patients treated with amlodipine [32]. The reduction in NT-proBNP level after amlodipine may be an indicator of the efficacy of amlodipine in improving cardiac function in patients with COPD with SPH [32]. One earlier study has also shown that amlodipine is an effective pulmonary vasodilator and effective in lowering PAP in patients with COPD and SPH [33]. In the current study, patients taking amlodipine had a lower RR during exercise, which might be possible due to a PAP-lowering effect during exercise. However, studies with noninvasive measurements (such as echocardiography) or invasive measurements (such as pulmonary artery catheters) during exercise to assess PAP should be designed to define this hypothesis.

In the current study, BW was higher in patients taking amlodipine than in those not taking amlodipine. Amlodipine is commonly associated with vasodilatory edema, which has been observed in 15.6% of the patients taking amlodipine [34]. Vasodectatic edema results from preferential arteriolar or precapillary dilation with fluid retention [34]. In the current study,

these patients did not appear to have apparent leg edema; however, their BW was still higher than that of patients not taking amlodipine.

There are limited studies on the effects of amlodipine on cardiorespiratory function in patients with COPD. Saikov et al. al showed that amlodipine has no effect on lung function and SpO$_2$ in COPD patients, [33] which is similar to our findings. There are no studies on the effect of amlodipine on the cardiorespiratory response during exercise in patients with COPD. This study is the first study to reveal that amlodipine did not affect exercise capacity, CI, WE, and HRQL in patients with COPD. A study of amlodipine in patients with heart failure also showed that amlodipine did not appear to affect exercise tolerance, symptoms and HRQL [35].

## Clinical implication

Cardiopulmonary responses and symptoms during exercise are important in patients with COPD. ß-blockers and amlodipine are common antihypertensive drugs used for patients with COPD and hypertension. According to the current study and published studies, ß-blockers did not show negative effects on lung function or cardiac function during exercise. Although there was a small but analytically significant decrease in SpO$_2$ and greater leg fatigue during exercise in patients taking bisoprolol, their exercise capacity and HRQL were not significantly different from those of patients not using bisoprolol. Patients taking amlodipine had a lower BP at rest, indicating its effect in controlling BP. In addition, a lower RR during exercise was noted in patients taking amlodipine. Previous studies have shown that amlodipine is effective in lowering pulmonary hypertension during exercise. These findings imply that the current practice of using bisoprolol and amlodipine has no prominent negative effects on cardiopulmonary responses during exercise. The use of amlodipine seemed to decrease the RR during exercise.

## Study limitations

This study had some limitations. First, this was a single-center study, and the number of cases was relatively small. Therefore, analytical bias could not be excluded. Multicenter studies with larger sample sizes are warranted. Second, this was a retrospective study. Therefore, prospective, randomized controlled studies should be conducted. Third, we did not measure dynamic hyperinflation or PAP during the exercise. The effects of β-blockers and amlodipine on the cardiopulmonary response to exercise were inferred from previous studies. Therefore, measurements of dynamic hyperinflation and PAP are necessary. Fourth, smoking is important factors to contributors to PAD, that often associated with leg pain and limited function [29]. However, we did not investigate the prevalence of PAD in these patients, and the impact of PAD could not be clearly defined. Further prospective studies could be performed to investigate the effects of amlopidine and bisoprolol on PAD and leg pain. Besides, we aimed to explore the effects of amlodipine and bisoprolol on cardiopulmonary function in patients with COPD. Therefore, we excluded patients with cardiac diseases, since these diseases themselves also affect cardiorespiratory function. Therefore, the results of this study cannot be applied to patients with COPD and cardiac diseases. Following this study in which we established the effect of amlodipine and bisoprolol in pure COPD patients, it would be a good subject to investigate in COPD patients with known cardiac diseases.

## Conclusions

Patients with COPD, especially older adults, often have high BP. ß-blockers and amlodipine are common antihypertensive drugs used for patients with COPD with hypertension. The impact of ß-blockers and amlodipine on cardiopulmonary responses and symptoms during

exercise is an important issue in patients with COPD. Patients with COPD taking bisoprolol had lower $SpO_2$ and more leg fatigue during peak exercise. Patients taking amlodipine had higher BW, lower BP at rest, and lower RR during exercise than those not taking amlodipine. Exercise capacity, $V_T$, and CI during exercise were similar between patients with and without bisoprolol or amlodipine.

## Supporting information

**S1 Raw data.**
(XLS)

## Author Contributions

**Conceptualization:** Chou-Chin Lan, Po-Chun Hsieh.

**Data curation:** Chou-Chin Lan, Po-Chun Hsieh, I-Shiang Tzeng, Yao-Kuang Wu.

**Formal analysis:** Po-Chun Hsieh, I-Shiang Tzeng, Mei-Chen Yang, Chih-Wei Wu, Wen-Lin Su, Yao-Kuang Wu.

**Funding acquisition:** Chou-Chin Lan.

**Methodology:** Mei-Chen Yang, Chih-Wei Wu, Wen-Lin Su.

**Supervision:** Chou-Chin Lan.

**Writing – original draft:** Chou-Chin Lan, Yao-Kuang Wu.

**Writing – review & editing:** Chou-Chin Lan, Yao-Kuang Wu.

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
