## [Decision Letter · Decision Letter 0]

15 Feb 2023

PONE-D-23-02534Impact of Bisoprolol and Amlodipine on Cardiopulmonary Responses and Symptoms during Exercise in Patients with Chronic Obstructive Pulmonary DiseasePLOS ONE

Dear Dr. Lan,

Thank you for submitting your manuscript to PLOS ONE. After careful consideration, we feel that it has merit but does not fully meet PLOS ONE’s publication criteria as it currently stands. Therefore, we invite you to submit a revised version of the manuscript that addresses the points raised during the review process.

Please submit your revised manuscript by Apr 01 2023 11:59PM. If you will need more time than this to complete your revisions, please reply to this message or contact the journal office at plosone@plos.org. Please include the following items when submitting your revised manuscript:A rebuttal letter that responds to each point raised by the academic editor and reviewer(s). You should upload this letter as a separate file labeled 'Response to Reviewers'.A marked-up copy of your manuscript that highlights changes made to the original version. You should upload this as a separate file labeled 'Revised Manuscript with Track Changes'.An unmarked version of your revised paper without tracked changes. You should upload this as a separate file labeled 'Manuscript'.If applicable, we recommend that you deposit your laboratory protocols in protocols.io to enhance the reproducibility of your results. Protocols.io assigns your protocol its own identifier (DOI) so that it can be cited independently in the future. For instructions see: https://journals.plos.org/plosone/s/submission-guidelines#loc-laboratory-protocols. Additionally, PLOS ONE offers an option for publishing peer-reviewed Lab Protocol articles, which describe protocols hosted on protocols.io. Read more information on sharing protocols at https://plos.org/protocols?utm_medium=editorial-email&utm_source=authorletters&utm_campaign=protocols.

We look forward to receiving your revised manuscript.

Kind regards,

Kartikeya Rajdev, MD

Academic Editor

PLOS ONE

Journal Requirements:

"This study was supported by grants from the Taipei Tzu Chi Hospital (TCRD-TPE-111-RT-3(1/3)) and the Buddhist Tzu Chi Medical Foundation (TCMF-JCT 111-14). "

Reviewers' comments:

Reviewer's Responses to Questions

**Comments to the Author**

1. Is the manuscript technically sound, and do the data support the conclusions?

Reviewer #1: Partly

Reviewer #2: Yes

Reviewer #3: Yes

2. Has the statistical analysis been performed appropriately and rigorously? 

Reviewer #1: Yes

Reviewer #2: Yes

Reviewer #3: Yes

3. Have the authors made all data underlying the findings in their manuscript fully available?

Reviewer #1: Yes

Reviewer #2: Yes

Reviewer #3: No

4. Is the manuscript presented in an intelligible fashion and written in standard English?

Reviewer #1: Yes

Reviewer #2: Yes

Reviewer #3: Yes

5. Review Comments to the Author

Reviewer #1: 1. Abstract-

BG- May help to specify the prevalence of Htn in COPD (reflecting the magnitude of the problem).

R- Since leg fatigue is listed, smoking history (PAD) could be additional factor needing to be specified)

C- Avoid subjective description like "mildly lower."

2. Main Paper-

Excl Criteria- CAD/HF were excluded but no mention of PAD (can occur without diagnosis of CAD) and may have independent impact on leg fatigue.

Result/Discussion- Consider adding a metric- Rate pressure product, which has been a reliable indicator of myocardial O2 demand and should be easily calculated from HR BP that you already collected.

Consider opining if Sp02 being lower could be a factor of lower HR at peak exercise in BB group? may need to supply that information as well

Also, description with regards to COPD regimen and need for Beat adr Rx/ interaction of such Rx with BB etc should be elaborated/discussed.

The comment of Amlodipine's association with lower RR and hypothesis of reduction in PA pressure can't be made based on the data presented as this was not prospectively evaluated. You could say, this "might be possible" and either noninvasive (Echo) or invasive (with PA catheter) study could be designed etc.

Clincal Impression-

Limitation- Data regarding PAD/Claudication was missing and this may have an implication in evaluation of leg fatigue which might be confounded, especially in these type of patients with smoking history.

- Baseline COPD therapy should be presented in the baseline characteristics

- Should mention that only mild to moderate COPD patients were evaluated in this study (Gold 4-2 patients only), hence might not apply to such severe COPD patients

- Also, gender/legal sex information is missing, so difficult to apply to population in general

Reviewer #2: The authors present a single center study examining the effects of bisoprolol and amlodipine on cardiopulmonary responses to exercise in COPD patients. One of the limitations of the study is the small size, however it does address a very important clinical question relevant to day-to-day cardiac, pulmonary and medicine practice. The authors have provided good data substantiating the notion that these medications are safe in COPD patients with regards to cardiopulmonary exercise parameters. Another limitation is the exclusion of patients with CAD and CHF, two very important indications for beta blocker therapy in cardiac practice (these cardiac conditions frequently coexist with COPD and the question under investigation becomes very relevant in this setting). However, it would be a good question for investigation in future studies to investigate the effects of beta blockers in COPD patients with known cardiac disease.

Larger multi center studies are needed to substantiate the findings of this study.

Reviewer #3: Comments to author: Please address these minor issues listed

1. Provide reference to Line 4 in third paragraph under Introduction section "Amlodipine not only controls hypertension but also reduces adverse cardiovascular events".

2. In paragraph 4 under Introduction section, Line 6 has bisoprolol and amlodipine in parenthesis implying both as Beta blockers which is not correct. Please address this.

3. Under Methods section, please elaborate the enrollment process. It is not clear at this stage whether this a retrospective study vs prospective study. Later in the paper you have mentioned this as a retrospective study under the limitations. However, it is unclear at this level.

4. There is no mention of health related quality of life (HRQL) and how it is measured in the methods section. Please elaborate regarding the same.

5. Under results section, when you are reporting acute exacerbations (AE), please explain if this is measured as one time event during the cardiopulmonary exercise testing or is there a certain follow up period that this variable is assessed.

6. Under discussion section, please provide more evidence available for related studies on amlodipine.

7. Under discussion section, paragraph 1, line 8, please correct grammatical errors - "in patients without or

without ß-blockers"

8. Page 11: Please recheck this line for any errors "A previous study also showed that β-blockers did not

show meaningful differences in muscle strength 19 and exercise capacity"

Overall, the sample size of the study is too small, but as you mentioned, i believe it is a good start to look at objective data regarding cardiopulmonary parameters to assess use of these medications in COPD.

6. PLOS authors have the option to publish the peer review history of their article (what does this mean?). If published, this will include your full peer review and any attached files.

Reviewer #1: No

Reviewer #2: No

Reviewer #3: No

---

## [Author Response · Author response to Decision Letter 0]

3 Apr 2023

PONE-D-23-02534

Impact of Bisoprolol and Amlodipine on Cardiopulmonary Responses and Symptoms during Exercise in Patients with Chronic Obstructive Pulmonary Disease

PLOS ONE

Answer: We provide two versions of the manuscript, one with highlighted revisions and one without highlighting.

Answer: Our manuscript meets PLOS ONE's style.

"This study was supported by grants from the Taipei Tzu Chi Hospital (TCRD-TPE-111-RT-3(1/3)) and the Buddhist Tzu Chi Medical Foundation (TCMF-JCT 111-14). "

Answer: The funders had no role in study design, data collection and analysis, decision to publish, or preparation of the manuscript. We added this description in the manuscript and cover letter. 

Answer: Data Availability Statement: All relevant data are within the paper. We added this description. 

Answer: We have provided our data. 

Important: If there are ethical or legal restrictions to sharing your data publicly, please explain these restrictions in detail. Please see our guidelines for more information on what we consider unacceptable restrictions to publicly sharing data: http://journals.plos.org/plosone/s/data-availability#loc-unacceptable-data-access-restrictions. Note that it is not acceptable for the authors to be the sole named individuals responsible for ensuring data access. We will update your Data Availability statement to reflect the information you provide in your cover letter.

Answer: We have provided our data. There is no any ethical or legal restriction.

Answer: The corresponding author has provided the ORCID number. 

Answer: We have review the references and all the references are complete and correct. 

Reviewers' comments:

Reviewer's Responses to Questions

Comments to the Author

1. Is the manuscript technically sound, and do the data support the conclusions?

Reviewer #1: Partly

Reviewer #2: Yes

Reviewer #3: Yes

Answer: Thanks to all reviewers. We did our best to conduct the research and write the manuscript.

2. Has the statistical analysis been performed appropriately and rigorously?

Reviewer #1: Yes

Reviewer #2: Yes

Reviewer #3: Yes

Answer: The co-author, I-Shiang Tzeng is a professional professor of statistics. The statistics was performed appropriately and rigorously

3. Have the authors made all data underlying the findings in their manuscript fully available? The PLOS Data policy requires authors to make all data underlying the findings described in their manuscript fully available without restriction, with rare exception (please refer to the Data Availability Statement in the manuscript PDF file). The data should be provided as part of the manuscript or its supporting information, or deposited to a public repository. For example, in addition to summary statistics, the data points behind means, medians and variance measures should be available. If there are restrictions on publicly sharing data—e.g. participant privacy or use of data from a third party—those must be specified.

Reviewer #1: Yes

Reviewer #2: Yes

Reviewer #3: No

Answer: We have provided the data underlying the findings described in our manuscript fully available without restriction.

4. Is the manuscript presented in an intelligible fashion and written in standard English? PLOS ONE does not copyedit accepted manuscripts, so the language in submitted articles must be clear, correct, and unambiguous. Any typographical or grammatical errors should be corrected at revision, so please note any specific errors here.

Reviewer #1: Yes

Reviewer #2: Yes

Reviewer #3: Yes

Answer: We did our best to write the manuscript and have native English speakers edit our English writing.

5. Review Comments to the Author

Reviewer #1: 1. Abstract-

BG- May help to specify the prevalence of Htn in COPD (reflecting the magnitude of the problem).

R- Since leg fatigue is listed, smoking history (PAD) could be additional factor needing to be specified)

C- Avoid subjective description like "mildly lower."

Answer:

1. BG: We changed the description as “The prevalence of hypertension in COPD patients ranges from 39% to 51%, and β-blockers and amlodipine are commonly used drugs for these patients.” We have also added this description to the Introduction section of the manuscript. 

2. Smoking status were not different between patients with or without bisoprolol or amlodipine. We added this description in abstract and main paper. We also provided data in table 1. We are sorry that we did not survey the prevalence of peripheral arterial disease (PAD) in the current study. We described this in Limitation of study.

3. We corrected the description of "mildly lower". 

2. Main Paper-

Excl Criteria- CAD/HF were excluded but no mention of PAD (can occur without diagnosis of CAD) and may have independent impact on leg fatigue.

Answer: We are sorry that we did not survey the prevalence of peripheral arterial disease (PAD) in the current study. We added this issue in Limitation of study. We also address some descriptions about this issue in Discussion section. 

Result/Discussion- Consider adding a metric- Rate pressure product, which has been a reliable indicator of myocardial O2 demand and should be easily calculated from HR BP that you already collected.

Answer:

Pressure rate product is an indicator to determine the myocardial O2 demand.

The calculation formula is: Rate Pressure Product (RPP) = Heart Rate (HR) * Systolic Blood Pressure (SBP) In the revised manuscript, we provide the RPP at rest and during exercise. However, our data showed no difference in RPP at rest or during exercise in patients taking or not taking bisoprolol, amlodipine.

Consider opining if Sp02 being lower could be a factor of lower HR at peak exercise in BB group? may need to supply that information as well

Also, description with regards to COPD regimen and need for Beat adr Rx/ interaction of such Rx with BB etc should be elaborated/discussed.

Answer: Thank you for your comments.

1. In Table 3, we provide HR during peak exercise. However, HR at peak exercise did not differ between patients with and without bisoprolol. Lower SpO2 during peak exercise did not appear to be associated with HR at peak exercise.

2. We provided the inhaled medications, oral β agonist (procaterol) and theophylline in these patients. There were no differences of inhaled medications, oral procaterol and theophylline in patients with or without β-blockers.

The comment of Amlodipine's association with lower RR and hypothesis of reduction in PA pressure can't be made based on the data presented as this was not prospectively evaluated. You could say, this "might be possible" and either noninvasive (Echo) or invasive (with PA catheter) study could be designed etc.

Answer: Thank you for your comment. We described this issue as “In the current study, patients taking amlodipine had a lower RR during exercise, which might be possible due to a PAP-lowering effect during exercise. However, studies with noninvasive measurements (such as echocardiography) or invasive measurements (such as pulmonary artery catheters) during exercise should be designed to define this hypothesis.”

Clinical Impression-

Limitation- Data regarding PAD/Claudication was missing and this may have an implication in evaluation of leg fatigue which might be confounded, especially in these type of patients with smoking history.

Answer: Thank you for your comments. We added the description in the Limitation Section: “smoking is important factors to contributors to PAD, that often associated with leg pain and limited function. However, we did not investigate the prevalence of PAD in these patients, and the impact of PAD could not be clearly defined. Further prospective studies could be performed to investigate the effects of amlopidine and bisoprolol on PAD and leg pain.”

- Baseline COPD therapy should be presented in the baseline characteristics

Answer: Thank you for your comments. We provided the treatment including inhaled medication, oral procaterol and theophylline. 

- Should mention that only mild to moderate COPD patients were evaluated in this study (Gold 4-2 patients only), hence might not apply to such severe COPD patients

Answer: We added the description: most patients were moderate to severe COPD (GOLD stage II or III), this conclusion is therefore more suitable for such patients.

- Also, gender/legal sex information is missing, so difficult to apply to population in general

Answer: We provided the gender information in table 1. There was no significant differences of gender differences between patients with our without amlodipine or bisoprolol. We added this description in the revised manuscript.

Reviewer #2: The authors present a single center study examining the effects of bisoprolol and amlodipine on cardiopulmonary responses to exercise in COPD patients. One of the limitations of the study is the small size, however it does address a very important clinical question relevant to day-to-day cardiac, pulmonary and medicine practice. The authors have provided good data substantiating the notion that these medications are safe in COPD patients with regards to cardiopulmonary exercise parameters. Another limitation is the exclusion of patients with CAD and CHF, two very important indications for beta blocker therapy in cardiac practice (these cardiac conditions frequently coexist with COPD and the question under investigation becomes very relevant in this setting). However, it would be a good question for investigation in future studies to investigate the effects of beta blockers in COPD patients with known cardiac disease. Larger multi center studies are needed to substantiate the findings of this study.

Answer: Thank you for these brilliant comments. We did our best to provide good data and manuscript to demonstrate the safety of these drugs in terms of cardiorespiratory exercise parameters in COPD patients.

1. We agreed that this is a single center with small sample size. We descripted this in Limitation of Study. Multicenter studies with larger sample sizes are warranted. 

2. In this study, we aimed to explore the effects of amlodipine and β-blockers on cardiopulmonary function at rest and during exercise in patients with COPD. Therefore, we excluded patients with cardiovascular disease, since these diseases themselves also affect cardiorespiratory function. Following this study in which we established the effect of amlodipine and beta-blockers in pure COPD patients, it would be a good subject to investigate in COPD patients with known cardiac disease. We describe this in the Limitation of Study to arouse readers' interest to conduct related research.

Reviewer #3: Comments to author: Please address these minor issues listed

1. Provide reference to Line 4 in third paragraph under Introduction section "Amlodipine not only controls hypertension but also reduces adverse cardiovascular events".

Answer: We provided the reference, Amlodipine and Landmark Trials: A Review. J Cardiol and Cardiovasc Sciences. 2021;5(3):1-8.

2. In paragraph 4 under Introduction section, Line 6 has bisoprolol and amlodipine in parenthesis implying both as Beta blockers which is not correct. Please address this.

Answer: We are sorry for the careless mistake. We have corrected the sentence “we conducted this study to comprehensively assess the effects of ß-blockers (bisoprolol and amlodipine) …” to “we conducted this study to comprehensively assess the effects of ß-blockers (bisoprolol) and amlodipine …”.

3. Under Methods section, please elaborate the enrollment process. It is not clear at this stage whether this a retrospective study vs prospective study. Later in the paper you have mentioned this as a retrospective study under the limitations. However, it is unclear at this level.

Answer: This is a retrospective study. We analyzed patients with COPD who met inclusion and exclusion criteria and underwent CPET. A total of 81 COPD patients who met the inclusion and exclusion criteria and completed CPET. We addressed this in the Methods section. 

4. There is no mention of health related quality of life (HRQL) and how it is measured in the methods section. Please elaborate regarding the same.

Answer: We have provided the measurement of HRQL such as COPD assessment test (CAT) and modified medical research council (mMRC) dyspnea scale. 

5. Under results section, when you are reporting acute exacerbations (AE), please explain if this is measured as one time event during the cardiopulmonary exercise testing or is there a certain follow up period that this variable is assessed.

Answer: AE of COPD is a sudden worsening of symptoms in patients with COPD. Severe AE is defined as AEs requiring urgent medical attention, such as emergency department visits or hospitalizations. We recorded the number of severe AEs in patients in the year preceding the analysis. 

6. Under discussion section, please provide more evidence available for related studies on amlodipine.

Answer: We thank you for your suggestion. The published study about the effects of cardiopulmonary response of amlodipine in patients with COPD are quite limited. We did our best to describe this issue in the revised manuscript. There are limited studies on the effects of amlodipine on cardiorespiratory function in patients with COPD. Saikov et al. al showed that amlodipine has no effect on lung function and SpO2 in COPD patients, which is the same as our study [1]. There are no studies on the effect of amlodipine on the cardiorespiratory response during exercise in patients with COPD. This study is the first study to reveal that amlodipine did not affect exercise capacity, cardiac index, work efficiency, and HRQL in patients with COPD. A study of amlodipine in patients with heart failure also showed that amlodipine did not appear to affect exercise tolerance, symptoms and HRQL in these patients [2]. 

1. Sajkov D, Wang T, Frith PA, Bune AJ, Alpers JA, McEvoy RD: A comparison of two long-acting vasoselective calcium antagonists in pulmonary hypertension secondary to COPD. Chest 1997, 111(6):1622-1630.

2. Udelson JE, DeAbate CA, Berk M, Neuberg G, Packer M, Vijay NK, Gorwitt J, Smith WB, Kukin ML, LeJemtel T et al: Effects of amlodipine on exercise tolerance, quality of life, and left ventricular function in patients with heart failure from left ventricular systolic dysfunction. American heart journal 2000, 139(3):503-510.

7. Under discussion section, paragraph 1, line 8, please correct grammatical errors - "in patients without or without ß-blockers"

Answer: We are sorry for the careless mistake. We have corrected this mistake. 

8. Page 11: Please recheck this line for any errors "A previous study also showed that β-blockers did not show meaningful differences in muscle strength 19 and exercise capacity"

Answer: In this text, we wanted to describe that patients taking bisoprolol had more leg fatigue, but without decline in exercise capacity. We corrected this sentence as “Although, we found more leg fatigue during exercise, exercise capacity was similar in patients with or without bisoprolol. Therefore, patients taking bisoprolol had more leg fatigue, but without decline in exercise capacity.”

The description " A previous study also showed that β-blockers did not show meaningful differences in muscle strength and exercise capacity " is intended to provide prior studies showing that bisoprolol did not appear to reduce muscle strength and exercise capacity.

Overall, the sample size of the study is too small, but as you mentioned, i believe it is a good start to look at objective data regarding cardiopulmonary parameters to assess use of these medications in COPD.

Answer: Thank you for these brilliant comments. We did our best to present good data to demonstrate the safety of these drugs in terms of cardiorespiratory exercise parameters in COPD patients. We agreed that this is a single center with small sample size. We descripted this in Limitation of Study. Multicenter studies with larger sample sizes are warranted. 

6. PLOS authors have the option to publish the peer review history of their article (what does this mean?). If published, this will include your full peer review and any attached files.

Do you want your identity to be public for this peer review? For information about this choice, including consent withdrawal, please see our Privacy Policy.

Reviewer #1: No

Reviewer #2: No

Reviewer #3: No

Answer: We respect reviewer's opinion.

---

## [Decision Letter · Decision Letter 1]

15 May 2023

Impact of Bisoprolol and Amlodipine on Cardiopulmonary Responses and Symptoms during Exercise in Patients with Chronic Obstructive Pulmonary Disease

PONE-D-23-02534R1

Dear Dr. Wu,

We’re pleased to inform you that your manuscript has been judged scientifically suitable for publication and will be formally accepted for publication once it meets all outstanding technical requirements.

Kind regards,

Kartikeya Rajdev, MD

Academic Editor

PLOS ONE

Additional Editor Comments (optional):

Reviewers' comments:

Reviewer's Responses to Questions

**Comments to the Author**

1. If the authors have adequately addressed your comments raised in a previous round of review and you feel that this manuscript is now acceptable for publication, you may indicate that here to bypass the “Comments to the Author” section, enter your conflict of interest statement in the “Confidential to Editor” section, and submit your "Accept" recommendation.

Reviewer #2: All comments have been addressed

Reviewer #3: All comments have been addressed

2. Is the manuscript technically sound, and do the data support the conclusions?

Reviewer #2: Yes

Reviewer #3: Yes

3. Has the statistical analysis been performed appropriately and rigorously? 

Reviewer #2: Yes

Reviewer #3: Yes

4. Have the authors made all data underlying the findings in their manuscript fully available?

Reviewer #2: Yes

Reviewer #3: Yes

5. Is the manuscript presented in an intelligible fashion and written in standard English?

Reviewer #2: Yes

Reviewer #3: Yes

6. Review Comments to the Author

Reviewer #2: I appreciate the authors taking the time to address my comments. Overall, I believe the comments have been adequately addressed by the authors at this point. They have clarified that the study was directed at pure COPD patients only and patients with known CV co-morbidities were excluded.

As long as comments from other reviewers have been met, the study appears suitable for publication at this point.

Reviewer #3: Authors have made all the changes and addressed all the comments as recommended by reviewers.

Overall as stated before the sample size of the study is too small and is one of the major limitations but as authors have stated, it is a good start to look at objective data regarding cardiopulmonary parameters to assess use of these medications in COPD.

7. PLOS authors have the option to publish the peer review history of their article (what does this mean?). If published, this will include your full peer review and any attached files.

Reviewer #2: No

Reviewer #3: **Yes: **Navya Alugubelli

---

## [Editor Report · Acceptance letter]

22 May 2023

PONE-D-23-02534R1 

Impact of Bisoprolol and Amlodipine on Cardiopulmonary Responses and Symptoms during Exercise in Patients with Chronic Obstructive Pulmonary Disease 

Dear Dr. Wu:

I'm pleased to inform you that your manuscript has been deemed suitable for publication in PLOS ONE. Congratulations! Your manuscript is now with our production department. 

Kind regards, 

on behalf of

Dr. Kartikeya Rajdev 

Academic Editor

PLOS ONE